# Genetic association analysis of human median voice pitch identifies a common locus for tonal and non-tonal languages
Yazheng Di [1,2], Joel Mefford [3], Elior Rahmani [4], Jinhan Wang[5], Vijay Ravi[5], Aditya Gorla[6], Abeer Alwan[5], Tingshao Zhu[1,2] & Jonathan Flint [7] ✉

The genetic influence on human vocal pitch in tonal and non-tonal languages remains largely unknown. In tonal languages, such as Mandarin Chinese, pitch changes differentiate word meanings, whereas in non-tonal languages, such as Icelandic, pitch is used to convey intonation. We addressed this question by searching for genetic associations with interindividual variation in median pitch in a Chinese major depression case-control cohort and compared our results with a genome-wide association study from Iceland. The same genetic variant, rs11046212-T in an intron of the *ABCC9* gene, was one of the most strongly associated loci with median pitch in both samples. Our meta-analysis revealed four genome-wide significant hits, including two novel associations. The discovery of genetic variants influencing vocal pitch across both tonal and non-tonal languages suggests the possibility of a common genetic contribution to the human vocal system shared in two distinct populations with languages that differ in tonality (Icelandic and Mandarin).

Human speech production is a complex process involving not only the coordinated activity of various organs[1], but also the social and cultural environments in which people learn to speak. As with all complex physiological processes, genetic effects likely play a role, but their extent and their molecular basis are largely unknown. Most prior molecular genetic studies have focused on disorders of language in a broader sense, such as disorders of reading and writing and developmental speech and language impairments[2,3], work that led to the identification of mutations in single genes that cause speech disorders[4]. Far fewer studies have looked at the genetic basis of speech production with acoustic measures[5], and it is unclear to what extent, if any, phonation characteristics are learnt rather than inherited (or influenced by the interplay of both). Finding the molecular basis of speech acoustics could shed light on this key, uniquely human, attribute.

There is a broad set of acoustic measures, each representing a specific aspect of the human vocal system and its psychological correlates[6]. A recent genome-wide association study (GWAS) discovered the first genetic locus associated with median voice pitch[6], which reflects the rate of vocal fold vibration and is perceived as how deep or high the voice sounds. By

recording the voice of 12,901 Icelanders during a reading task, a single locus was found that exceeded a corrected significance threshold, on chromosome 12 at the adenosine triphosphate binding cassette subfamily C member 9 gene (*ABCC9*).

The Icelandic finding, so far not replicated in an independent sample, raises several questions: First, does the chromosome 12 locus have the same effect in other populations? Iceland is genetically and linguistically homogeneous with minor dialectal variation in their non-tonal language[6]. Second, how does the chromosome 12 locus affect pitch in speakers of tonal languages? Pitch often represents word emphasis and speakers' emotional context and can convey semantic information, especially in tonal (or pitch-accented) languages, where pitch is used to differentiate word meanings[7]. Around 60–70% of the world's languages are tonal languages[7]. Third, does the effect exist in spontaneous speech, or is it restricted to a reading task, like the one used in the Icelandic study? Finally, given reported differences in pitch attributable to variation in mood[8–10], to what extent do the findings depend on the mood of the subjects? To address these questions, we performed GWAS on voice pitch measured in 7654 Han Chinese Women, recruited for a case-control genetic study of the origins of major depressive

[1]CAS Key Laboratory of Behavioral Science, Institute of Psychology, Beijing 100101, China. [2]Department of Psychology, University of Chinese Academy of Sciences, Beijing 100049, China. [3]Department of Neurology, University of California Los Angeles, Los Angeles, CA, USA. [4]Department of Computational Medicine, University of California Los Angeles, Los Angeles, CA, USA. [5]Department of Electrical and Computer Engineering, University of California Los Angeles, Los Angeles, CA, USA. [6]Bioinformatics Interdepartmental Program, University of California Los Angeles, Los Angeles, CA, USA. [7]Department of Psychiatry and Biobehavioral Sciences, Brain Research Institute, University of California Los Angeles, Los Angeles, CA, USA. ✉e-mail: JFlint@mednet.ucla.edu

disorder (MDD). The design allowed us to incorporate changes in mood into our analysis and to explore whether genetic effects on pitch were the same or different in tonal and non-tonal languages.

## Results

About 364,929 voice segments were manually identified from 7654 subjects (3641 cases and 4013 controls). The mean duration of audio extracted from case interviews was 192.73 s (SD = 196.68), approximately twice as long as for controls (97.44 s, SD = 122.96). Segments were manually classified for their noise level and accent. 60% of the subjects spoke in standard Mandarin, whereas the rest spoke either their local languages or Mandarin with local accents. 78% of the interviews were recorded with no or low noise levels. The voice and demographic information in the case/control subgroups are reported in Supplementary Table 1.

We used the median F0 to measure pitch, the same measure used in the Iceland study[6]. The mean over all subjects was 206.25 Hz (SD = 26.31 Hz). Pitch was associated with age ($\beta = -0.25$, $P = 9.65 \times 10^{-108}$), and, after correcting for the effects of the collection site, weakly associated with MDD $\beta = 0.14$, $P = 0.013$. After adjusting for age and MDD, pitch was significantly associated with height, BMI, education level, and other confounding variables listed in Supplementary Table 2.

The heritability of pitch was 20% (95% CI: 10 to 29%, close to the estimation in the Iceland study: 17%, 95%CI: 9 to 24%) after adjusting for MDD and other covariates as listed in Supplementary Table 2. The genetic correlation of pitch between MDD cases and controls was not significantly different from 1 (rg = 1.00, SE = 0.43, P = 0.5). However, we conservatively performed GWAS on the two cases and controls separately and then combined the effects using meta-analysis. No genome-wide significant loci were identified (Fig. 1). The variant rs11046212-T in an intron of the *ABCC9* gene, associated with a pitch in the Iceland study[6], was one of the strongest associations in our study ($\beta = 0.09SD$, 2.3 Hz per allele, $P = 2.33 \times 10^{-7}$). Association with one other locus on the same chromosome, rs10859172-C, achieved almost equal significance ($\beta = -0.08SD$, $P = 2.06 \times 10^{-7}$). We tested for heterogeneity and found that the effects of these two SNPs were not heterogeneous between cases and controls (rs11046212, heterogeneity P value = 0.61; rs10859172, heterogeneity P value = 0.83).

We explored the association with two other measures of pitch, the first and the third quartile of F0 (see Supplementary Note 1, Supplementary Figs. 1–3, and Supplementary Table 3 for a comparison of different quantiles of F0.). Two loci, one associated with the first and one associated with the third quartile of F0, achieved genome-wide significance (Supplementary

Table 4). They were rs11046212-T with the first quartile of F0 ($\beta = 0.10SD$, $P = 3.32 \times 10^{-8}$), and rs10859172-C with the third quartile of F0 ($\beta = -0.09SD$, $P = 2.53 \times 10^{-8}$). Since we observed variation between hospitals in the distribution of voice features (Supplementary Table 5), we also ran a hospital-level meta-analysis and confirmed that the associations between SNPs and pitch were not heterogeneous between hospitals (Supplementary Note 2).

We compared several F0 statistics in our study with the 7278 females in the Iceland study. The results are shown in Supplementary Table 6. The pitch (F0 median) was significantly higher in the Chinese sample than in Iceland (t-statistic= 8.31, $P = 1.06 \times 10^{-16}$). Pitch in our dataset varied more within-person than in the Iceland study (standard deviations of F0: t-statistic= 56.17, $P<1.23 \times 10^{-308}$), and we observed higher skewness in the distribution of F0 in Chinese individuals (t-statistic = 32.97, $P = 3.14 \times 10^{-230}$) which we attribute to the greater use pitch plays in conveying meaning in Chinese tonal languages than in Icelandic.

We then built a polygenic score (PGS) based on several different *P* value thresholds (PT) from the summary statistics provided by the Iceland study and tested the predictive performance of our data. The results are shown in Fig. 2. The best fit included only five SNPs at PT = $2.30 \times 10^{-7}$, and explained 0.61% of the variance in pitch ($P = 3.32 \times 10^{-8}$). The PRS model explained a proportion of variance in the case/control subgroups that was similar to that in the entire group. (Supplementary Figs. 4, 5).

We evaluated whether the observed fraction of results displaying the same direction of allelic effects across studies was significantly greater than expected by chance (that is, 50%) using binomial sign tests. Table 1 gives the number of LD-independent SNPs in the Iceland study at a set of *P* value thresholds, and the fraction of these SNPs displaying the same direction of effect in the Chinese group and a one-sided binomial test *P* value. 98.99% of the SNPs showed the same direction at a *P* value threshold <$1 \times 10^{-6}$ (One-sided binomial $P = 1.58 \times 10^{-28}$).

Finally, we performed a meta-GWAS combining the Iceland study and our case/control subgroups. The Manhattan plot is shown in Fig. 3. We found four genome-wide significant hits, of which two were novel. The four top hits from the cross-population meta-analysis, together with the rs10859172 found in our study, are listed in Table 2. The most significant association was for the rs11046212-T at the *ABCC9* locus ($P = 7.50 \times 10^{-24}$), which showed a similar effect in Chinese ($\beta = 0.09$ SD, 2.3 Hz per allele) and Icelanders ($\beta = 0.11$ SD, 2.1 Hz per allele). Its allele frequency, however, was lower in China (0.25) than in Iceland (0.48). Results of all SNPs that achieved genome-wide significance are in Supplementary Data 1.

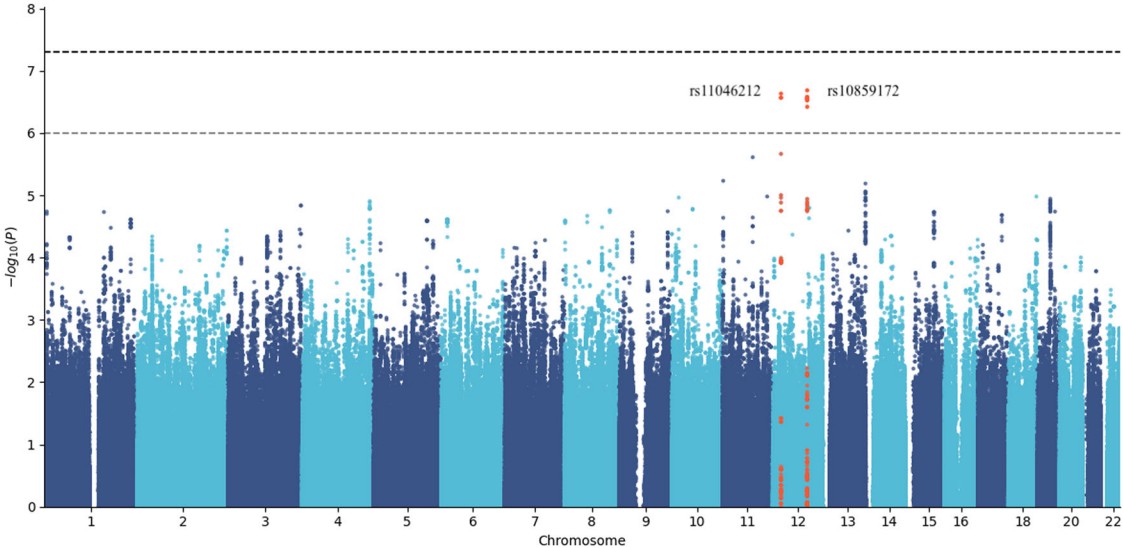

**Fig. 1 | Manhattan plot of the association results for voice pitch (median F0) in 7654 Han Chinese women.** The dashed horizontal lines indicate genome-wide significance (top, $5 \times 10^{-8}$) and suggestive significance (bottom, $1 \times 10^{-6}$).

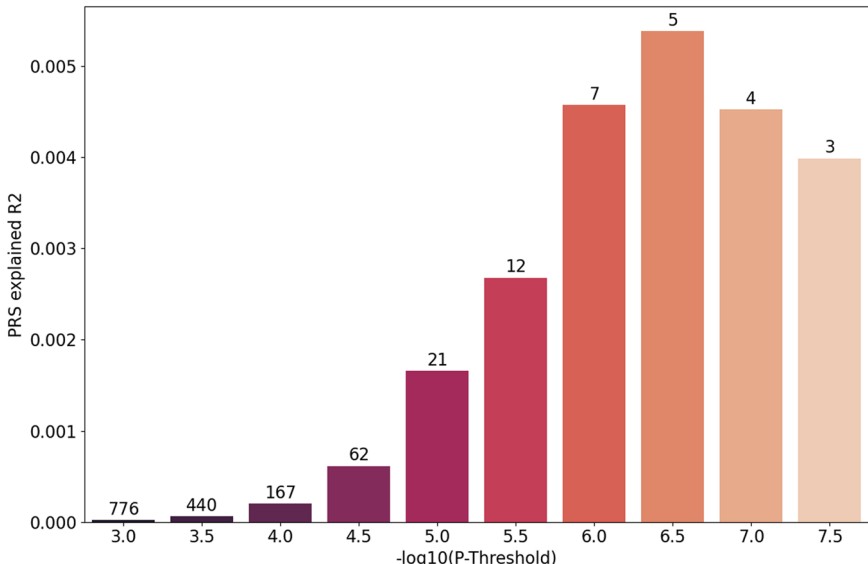

**Fig. 2 | The predictive performance of polygenic scores based on the Iceland study.** The number of SNPs is labeled on each bar.

## Table 1 | Binomial sign test

| P_threshold | N_total | N_positive | fraction | Binomial_P |
|---|---|---|---|---|
| 5.00E-08 | 20 | 20 | 100.00% | 9.54E-07 |
| 1.00E-07 | 21 | 21 | 100.00% | 4.77E-07 |
| 1.00E-06 | 99 | 98 | 98.99% | 1.58E-28 |
| 1.00E-05 | 133 | 120 | 90.23% | 3.67E-23 |

For varying $P$ value thresholds ($P\_threshold$), the total number of SNPs ($N\_total$), number of SNPs showing consistent direction of effect ($N\_positive$), fraction of these with consistent direction of effect, and significance of a one-sided binomial test ($Binomial\_P$).

Cross-ancestry fine-mapping analysis of the *ABCC9* locus identified four variants as probable causal variants in the 95% credible set (Supplementary Table 7). The variant rs11046212 had the maximum posterior probability (0.35), which was inferred as causal in both EUR and EAS populations (population-specific causal probability >0.99).

## Discussion

We were able to answer three questions about the genetic basis of pitch production. First, we established that at least some of the same genetic loci are present in both Chinese and Icelandic populations. The loci we identified contribute to voice pitch in both tonal and non-tonal languages. We have no evidence that the loci differ in direction or size of the effect, despite the marked differences in the structure of the languages and the semantic use of pitch in Mandarin. Second, the effects are found in both spontaneous speech as well as in reading tasks, revealing persistent genetic effects on voice pitch across different contexts. Third, they are not dependent on mood: a heterogeneity test was not significant, suggesting consistent genetic effects between MDD patients and healthy people.

What does the finding of common genetic underpinnings in Mandarin Chinese (a language in which word meaning is conveyed by variation in pitch) and Icelandic (a language in which pitch does not play this role) reveal about the biology of speech and language? Marked differences in pitch patterns between tonal and non-tonal languages have been demonstrated in previous studies[11,12], yet we found a cross-linguistic consistency in the influence of *ABCC9* locus on pitch. We also observed a high degree of consistency in the direction of the genetic effects on pitch, with 98.99% of the SNPs exhibiting effects in the same direction at a $P$ value threshold $<1 \times 10^{-6}$. We think this consistency is because the analyzed phenotype, median F0, is primarily related to the non-linguistic production of speech, rather than to

meaning. Spoken language, which involves cognition and emotional process, is more likely to be reflected in the variation of F0 over time during speech[13], not to the mean or median values. Indeed, we found that it was the changes in pitch, not pitch itself, that were genetically correlated with MDD[10].

Voice pitch is primarily modulated by fine changes in the tension of the vocal folds, which is mainly achieved by flexing the cricothyroid muscle, which causes the thyroid cartilage to tilt relative to the cricoid cartilage, thereby stretching the vocal folds[11]. Greater tension causes the vocal folds to vibrate at a higher frequency during voicing, producing a higher-pitch sound. The *ABCC9* gene may influence voice pitch through hormonal pathways related to adrenal gland steroids, which produce several steroids known to influence voice pitch[6,14]. It can, alternatively, exert through non-hormonal mechanisms, affecting proteins in muscles related to the vocal fold or vocal tract[6].

There are several limitations to our study. First, only women were included in the CONVERGE analysis. Although the Iceland study indicated that the effects of *ABCC9* are irrespective of sex, it is unknown whether this finding is applicable to Chinese populations. Second, our analysis focused solely on mapping median F0. Other features, such as the variability of F0 and vowel acoustics, which represent different and possibly more crucial aspects of human vocal control ability, remain unexplored in our study.

In summary, through GWAS on Chinese women, we replicated the effects of a genetic locus at the *ABCC9* gene on voice pitch. In combination with the Iceland study, we found two novel hits for pitch. These findings revealed common genetic effects on pitch across populations and languages.

## Methods
### Participants

The sample included 7654 women recruited from 55 provincial mental health centers in China as part of the CONVERGE (China, Oxford, and VCU Experimental Research on Genetic Epidemiology) study. Cases were aged between 30–60, with ≥ two episodes of MDD that met the DSM-IV criteria[15]. Control subjects, screened to exclude a history of MDD, were recruited from patients undergoing minor surgical procedures at general hospitals and individuals attending local community centers. Sample collection is described in detail in earlier work[16–18]. This study was approved by the Ethical Review Board of Oxford University (Oxford Tropical Research Ethics Committee) and local hospital review boards. All participants provided written informed consent.

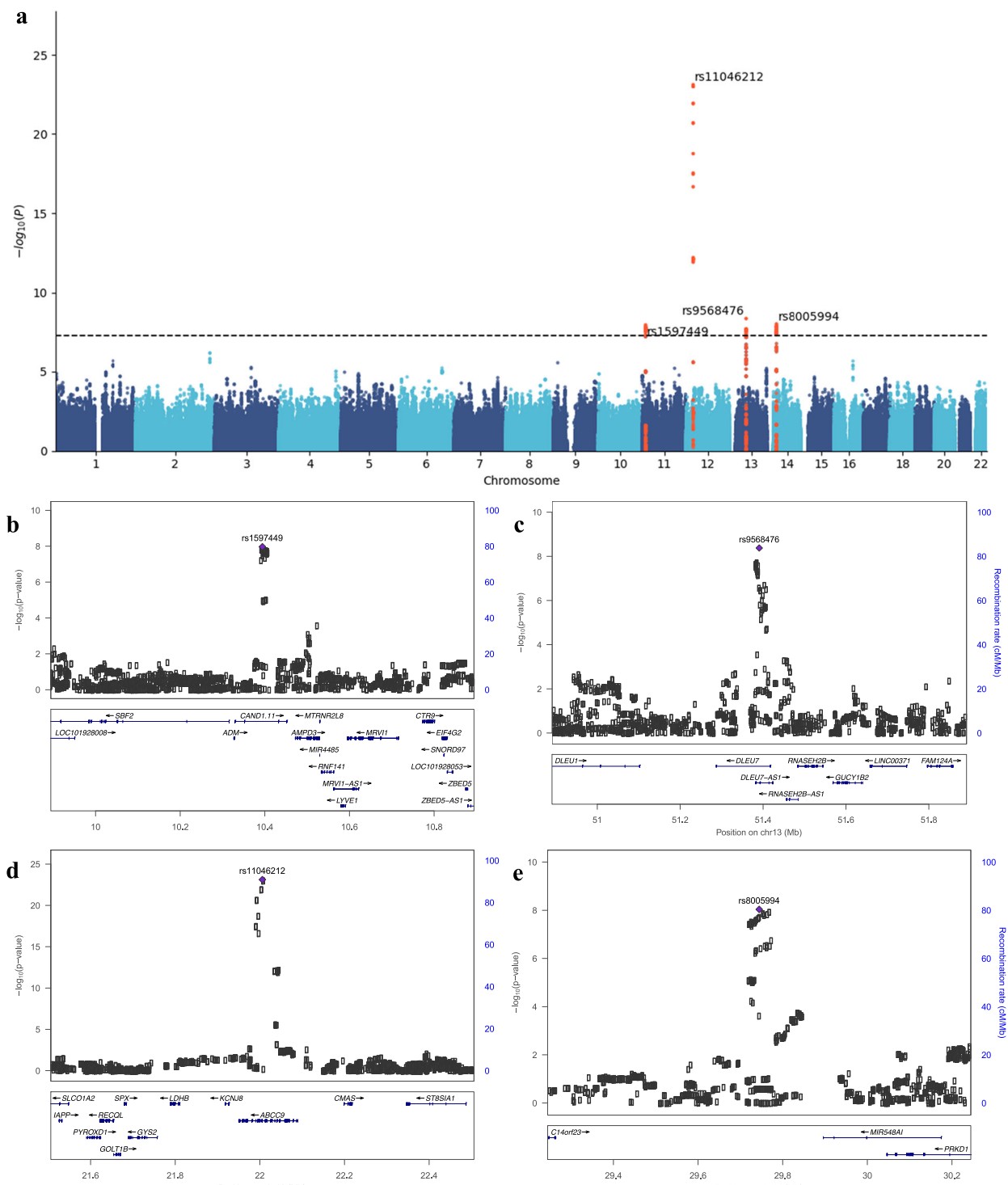

**Fig. 3 | Four loci associated with voice pitch (median F0). a** Manhattan plot of the cross-population meta-GWAS. The dashed horizontal line indicates genome-wide significance (top, $5 \times 10^{-8}$). Novel hits are marked in bold. **b–e** Regional plots of the top variants associated with voice pitch. The $-\log10(P$ value) of imputed SNPs associated with pitch is shown on the left y-axis. The recombination rates expressed in centimorgans (cM) per Mb (GRCh37; blue lines), are shown on the right y-axis. Position in Mb is on the x-axis. The plots were drawn using LocusZoom[31].

## Data collection

The voice data were obtained from semi-structured interviews, where the conversations between the subjects and the interviewer were recorded in hospital interview rooms. The interview was designed to assess psychiatric and demographic information in cases and controls. The length of case interviews was approximately three times longer than for controls because of a more detailed assessment of psychopathology and MDD risk factors

(details of the interview protocol are provided in Supplementary Note 3). The recordings were not standardized and varied in quality and content. All recordings were listened to, and segments that contained only the patient's voice at an adequate quality for analyses were identified. During this procedure, segments were manually labeled as to whether the speaker used a local dialect or had a non-standard accent for Mandarin Chinese, and a

**Table 2 | Top SNPs associated with pitch (median F0) in cross-population meta-analysis and in our study**

| SNP | Chr | POS | EA | OA | Gene | Meta-analysis | | Chinese (N = 7654) | | | | Icelandic (N = 12,901) | | | |
|---|---|---|---|---|---|---|---|---|---|---|---|---|---|---|---|
| | | | | | | P | Het_P | Beta | 95% CI | P | AF | Beta | 95% CI | P | AF |
| rs1597449 | 11 | 10394222 | T | A | CAND1.11 | 1.07E-08 | 0.06 | 0.05 | (0.01,0.08) | 6.20E-03 | 0.48 | 0.07 | (0.04,0.09) | 3.20E-07 | 0.47 |
| rs11046212 | 12 | 22006116 | T | C | ABCC9 | 7.50E-24 | 0.41 | 0.09 | (0.06,0.12) | 2.33E-07 | 0.25 | 0.11 | (0.09,0.14) | 2.60E-18 | 0.48 |
| rs956476 | 13 | 51390842 | C | A | DLEU7 | 4.18E-09 | 0.26 | −0.04 | (−0.07,−0.01) | 0.02 | 0.48 | −0.07 | (−0.10,−0.05) | 2.60E-08 | 0.58 |
| rs8005994 | 14 | 29744532 | A | G | LOC102724934 | 9.19E-09 | 0.47 | −0.04 | (−0.07,−0.01) | 9.91E-03 | 0.42 | −0.07 | (−0.10,−0.05) | 1.40E-07 | 0.65 |
| rs10859172 | 12 | 92028791 | C | T | LOC105369896 | 1.74E-04 | 0.001 | −0.08 | (−0.11,−0.05) | 2.06E-07 | 0.43 | −0.01 | (−0.04,0.02) | 0.46 | 0.55 |

The first six columns are rsid (SNP), chromosome (Chr), genomic position (Pos) on Human Genome Reference GRCh37.p5, effect allele (EA), other allele (OA), and gene. The next two columns are the results of cross-population meta-analysis: P values of association and P values for the heterogeneity test. The next four columns are GWAS results of our Chinese cohort: effect size (Beta), 95% confidence interval (CI), P value, and effect allele frequency (AF). The last four columns are the results of the Icelandic study: effect size (Beta), 95% confidence interval (CI), P value, and allele frequency (AF).

number was assigned to each recording to represent the level of background noise (four levels: 1. No noise; 2. Low noise; 3. Mild noise; 4. High noise). All segments from the same subject were concatenated into one, and down-sampled to 8 kHz. Two postgraduate psychological students listened to all segments to ensure that no speech voice other than the interviewed subjects was included. All participants provided DNA samples for genetic analysis.

DNA was extracted from saliva samples using the Oragene protocol. Genotypes were acquired from low-coverage sequencing data from which SNPs were imputed. A sensitivity threshold of 90% to SNPs in the 1000 G Phase1 ASN panel was applied for SNP selection for imputation. Genotype likelihoods were calculated using a sample-specific binomial mixture model implemented in SNPtools (version 1.0)[19], and imputation was performed using BEAGLE (version 3.3.2)[20]. A second round of imputation was performed with BEAGLE at biallelic SNPs polymorphic in the 1000 G Phase 1 ASN panel using the 1000 G Phase 1 ASN haplotypes as a reference panel. A final set of allele dosages and genotype probabilities was generated from these two datasets by replacing the results in the former with those in the latter at all sites imputed in the latter. We applied a conservative set of inclusion thresholds for SNPs for genome-wide association study: (a) $p$ value for violation Hardy–Weinberg equilibrium $p > 10\text{-}6$, (b) Information score $p > 0.9$, (c) minor allele frequency $>0.5\%$. Full details of the method and results are given in ref. 16.

### Voice pitch calculation

We used the median F0 to measure pitch, which is more robust to outliers than mean values and is the same measure used in the Iceland study[6]. Given an audio segment, a time series of F0 values was computed using the Sub-harmonic Summation method and subsequently smoothed[21]. However, as our voice data were from spontaneous speech and contained more phonetic variation, we also calculated the first and the third quartiles of the F0 series, which were used for sensitivity tests. The calculation was implemented using the openSMILE package v2.4.2[22].

### Heritability, genetic correlation, GWAS, and meta-analysis

The SNP-based heritability estimation used a GREML (generalized restricted maximum likelihood) method. GWAS was performed using a mixed model linear regression. Both were implemented in LDAK[23]. We adjusted for age, height, BMI, education level, marital status, occupation, social class, accent, noise level, total audio durations, the total number of audio segments concatenated, and 20 genetic principal components (PCs). The genetic correlation of pitch between cases and controls was calculated based on the individual genotype data, using the bivariate GREML method implemented in GCTA[24].

Because of the expected differences in speech between the cases and controls[10], we performed GWAS in the case and control subgroups separately. Then, we used meta-analyses to combine the two results. For cross-population analysis, we performed a meta-analysis of the summary statistics provided by the Iceland study, GWAS of our case subgroup, and GWAS of our control subgroup. The meta-GWAS were implemented in Metal[25]. Cochran's Q-test[26] was implemented for the heterogeneity test. The summary statistics of the Iceland study[6] were lifted to GRCh37/hg19 and matched with our genotype data.

### Polygenic score and binomial sign tests

We used the SNP associations from the Iceland study[6] to construct PGS in the Chinese cohort. We first performed LD-based clumping (pairwise $r^2 > 0.5$ in Chinese, 50 kb window) to remove markers from highly correlated SNP pairs. Then, we constructed PGS based on varying $P$ value thresholds from $5 \times 10^{-8}$ to 0.001 with an interval of $5 \times 10^{-8}$. We assessed the predictive value of polygenic scores in the Chinese cohort by linear regression, with adjustment for the same covariates used in the GWAS analyses.

Using the same sets of SNPs and the $P$ value thresholds ranging from $5 \times 10^{-8}$ to 1 in the Iceland study, we applied a binomial sign test to determine whether the number of SNPs demonstrating consistent

directions of allelic effects between Icelandic and Chinese was greater than expected by chance (that is, a one-sided test of whether this fraction is greater than 0.5).

## Cross-ancestry fine-mapping

We used SuSiEx[27,28] for fine-mapping analysis, which required two separate GWAS summary statistics in the single population and the corresponding LD matrixes. All variants in the *ABCC9* locus that achieved genome-wide significance in the meta-analysis (Supplementary Data 1) were included. The LD matrixes were calculated using 1000 Genomes Project[29] EAS/EUR samples as reference panel.

## Statistics and reproducibility

We used the median F0 to measure pitch from 7654 Han Chinese Women. Saliva samples were collected and DNA was extracted using the Oragene protocol, as described above. GWAS on pitch was conducted in MDD cases and controls separately using a linear mixed model implemented in LDAK[23], and then combined using a meta-analysis tool Metal[25] (2011-03-25 version). The cross-population GWAS was conducted through a meta-analysis by combining the Chinese MDD cases, controls, and the Icelandic summary statistics, while the Icelandic results were lifted to GRCh37/hg19 and matched with the CONVERGE genotype data. Cross-population fine-mapping was implemented in SuSiEx[27,28]. The polygenic scores were calculated using a $P$ value-based thresholding and clumping method implemented in PRSice[30]. In the one-sided binomial sign tests, we defined the null hypothesis as the probability of observing a consistent direction of genetic effects (as outlined in Table 1 for the total number of SNPs) being equal to 0.5. Conversely, the alternative hypothesis suggested that this probability is greater than 0.5.

## Reporting summary

Further information on research design is available in the Nature Portfolio Reporting Summary linked to this article.

## Data availability

The GWAS summary statistics of the median pitch in Chinese and the meta-GWAS are publicly available at FigShare (https://doi.org/10.6084/m9.figshare.24995963). The GWAS summary data for the median pitch in the Iceland study are available at Zenodo (https://doi.org/10.5281/zenodo.7152461). Due to the sensitive nature of the raw audio files and in adherence to privacy considerations, these files cannot be made publicly available. However, we are committed to facilitating scientific progress and transparency. Thus, secondary data derived from these audio files, specifically voice features, are available upon reasonable request. Researchers interested in accessing these data should contact the corresponding author, Jonathan Flint at JFlint@mednet.ucla.edu.

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

## Acknowledgements

This work was funded by NIH grant MH-122596.

## Author contributions

J.F. conceived and organized this study. T.Z. and A.A. were responsible for overseeing the collection, cleaning, and analysis of the voice data. Y.D., J.W., V.R. and A.G. contributed to the voice data analysis. Y.D., J.M. and E.R. contributed to the genetic analysis. Y.D. and J.F. drafted the manuscript. All authors revised and approved the final version of the manuscript.

## Competing interests

The authors declare no competing interests.
