## [Peer Review File · Communications Biology]

Reviewers' comments:

Reviewer #1 (Remarks to the Author):

Di et al. have performed a GWAS of voice pitch in 7,654 Han Chinese Women from the CONVERGE study, originally recruited for the study of major depression and replicated the top signal near the ABCC9 gene discovered in a previous GWAS of voice pitch based on Icelandic individuals. Despite the stark differences in the language, social and cultural features between the two populations, the ABCC9 variant showed significant associations with impressively similar effect sizes in both studies. A polygenic score based on the deCODE study predicts the F0 voice pitch phenotype significantly in the CONVERGE study suggesting a high genetic overlap. The authors further meta-analyze the GWAS results from both studies and report in total four GWAS hits of which two are novel.

This is an important study that independently replicates an important finding from the deCODE study which was a first of its kind. It's great to see that the CONVERGE cohort is continuing to contribute to human genetics studies. The clear replication of the ABCC9 signal reflects the quality of the phenotype collected from the CONVERGE participants, which needs to be appreciated. I do not have any major concerns and recommend publishing this. But I have a few comments and suggestions, which I'll list below.

1. Phenotypically there is a strong association between MDD status and F0 voice pitch, almost as strong as the association with age, which makes one wonder if voice pitch has a value in predicting MDD. I later realized that in fact, that was a major goal of the study and the authors would want to report that in detail elsewhere as a separate paper. Fair enough. But, as a reader, I'd still like to know what is genetic correlation between MDD and F0 voice pitch is in the current study. Could the authors please report that here? Also, if possible, report the genetic correlation of voice pitch between cases and controls and heritability estimates separately in cases and controls. Finally, polygenic score associations with voice pitch separately in cases and controls and if there are any significant differences in the effect size between the two.
2. I am curious about the genetic correlation between CONVERGE and deCODE studies. Did the author try to estimate it?
3. The authors use a clumping and thresholding approach to analyze polygenic scores. Did they try using any newer methods such as LDpred or PRSCS? Just because the PRS based on the top variants showed the strongest association, I wouldn't conclude that the voice pitch phenotype is oligogenic. How much proportion of the heritability is explained by the top variants? Does that number justify the claim that voice pitch is an oligogenic trait?
4. Did the authors perform fine mapping of the ABCC9 locus? How many variants fall within the 95% credible set? Does meta-analyzing Chinese and Icelandic studies help narrow down the causal variants?
5. The authors avoided reporting the genetic correlation of voice pitch with other diseases and traits. Is there any specific reason? I'd assume that meta-analysis would provide more power to do such analysis, and the results would provide some interesting new insights into the environment and disease correlates of voice pitch in the population.
6. Did the authors explore stratified heritability analysis to identify tissues and cell types associated with voice pitch? With a heritability of 20% using LDSC, I'd assume that the sumstats would be well suited for

such an analysis.

7. The methods section doesn't describe anything about the genetic data of CONVERGE participants. It is important to add that information here, even though it may have been described elsewhere. Will the summary statistics made available publicly?

Reviewer #2 (Remarks to the Author):

This fascinating study investigates genetic influences in interindividual variation in median voice pitch during speech in a Chinese cohort of females. There have been very few papers applying state-of-the-art genomic approaches to normal variation in speech properties of this kind, so this is important work in an emerging area of considerable interest. Remarkably, the authors report some striking convergences with a prior study of median voice pitch during reading that had been carried out in an Icelandic sample. In particular, the clear signal for the identical SNP in the ABCC9 gene in these two completely independent systematic genome-wide screens is exciting and definitely of interest to others in the community and the wider field. The work is mostly convincing, and statistical analyses seem appropriate and valid as far as I can tell. The paper is generally well-written, clear and concise, and nicely presented. However, I do have reservations regarding the framing of the study, especially with respect to what the study can (or rather cannot) reveal about tonal/non-tonal language distinctions, and the biology of human speech more broadly, and would recommend more caution in the conclusions that the authors are drawing.

Specifically, my comments for consideration by authors are as follows:

1) The overarching framing of the study could be misleading (through the abstract and in several places in the introductory paragraphs of the paper) with potential to fuel misunderstandings by readers, especially for those who may be less familiar with speech/language sciences.

a. The abstract sets out the apparent scope of the study with the following: "The origins of tonal and non-tonal languages have long been a subject of linguistic inquiry. In tonal languages, such as Mandarin Chinese, pitch changes differentiate word meanings, whereas in non-tonal languages, such as Icelandic, pitch is used to convey intonation. We addressed this question by searching for genetic associations with variation in pitch..." (lines 21-24). Crucially, the phenotypic measure that is under investigation in the current study is the median fundamental (F0) frequency of an individual's voice i.e. the median natural pitch that a person uses when they are speaking. But the role of tone in linguistics concerns how a speaker uses pitch modulation within an utterance in relation to prosody (as in non-tonal languages) or to directly modify the meaning/inflection of words (as in tonal languages). A person's median F0, while certainly an interesting trait to investigate in its own right, is not so likely to be informative for resolving long-standing questions about "origins of tonal and non-tonal languages". A more meaningful target phenotype for a genetic study related to tonal/non-tonal distinctions would be a measure of a person's ability to modulate tone around their median pitch; notably, this avenue is not pursued within the current paper.

b. The above issue is compounded by some lack of clarity/precision in phrases like "genetic associations with variation in pitch" (line 24) and "the first genetic locus associated with variation in voice pitch" (lines 44-45). Such phrases do not make clear that "variation" here is meant as a group-level reference to interindividual variation of median pitch rather than a claim about identifying genetic associations with a

phenotype based on intraindividual variation (i.e. how tone is being modulated during utterances by each person in a cohort). It makes it open to misunderstanding by non-expert readers, who are likely to assume the latter. And nowhere in the title or abstract do the authors mention that the focus here is limited to a person's median pitch, which makes it even more likely that readers could (given the predominant tonal/non-tonal framing) jump to the wrong conclusions about the nature of the target trait.

c. In the final sentence of the abstract, the authors conclude that their findings show “a genetic contribution to a fundamental capability, the physiological basis for pitch control in speech, shared by all humans, regardless of their linguistic or cultural background” (lines 29-31). Given the restricted focus on interindividual variation in median F0, GWAS information from only two languages/populations, and without any investigations of how genetic factors influence the modulation of tone while speaking, it seems to be an overstatement to make such a broad claim about shared genetics of “pitch control in speech” across linguistic/cultural backgrounds. As far as I can see, that claim is not something that can be resolved with the current limited study design, and so I would recommend rewriting the abstract with a more measured and modest conclusion. In addition, some adjustment of the paper's title seems warranted to reflect these points and to avoid casual readers being misled over the broader significance of the findings for understanding of tonal/non-tonal distinctions.

d. To the credit of the authors, very close to the end of the manuscript, they do concede that the chosen trait for study here is not that informative for investigating tonal/non-tonal questions: “Marked differences in pitch patterns between tonal and non-tonal languages have been demonstrated in previous studies yet we found a cross-linguistic consistency in the influence of ABCC9 locus on pitch. We think this consistency is because the analyzed phenotype, median F0, is primarily related to the non-linguistic production of speech, rather than to meaning. Spoken language, which involves cognition and emotional process, is more likely to be reflected in the variation of F0 over time during speech, not to the mean or median values.” (Lines 149-155). I agree, and would suggest that the lack of informativity of median F0 with respect to deeper questions about biology of speech and language (including tonal/non-tonal distinctions) needs to be acknowledged upfront in the manuscript, and should be taken into account for the overall framing of the work, rather than being treated as a kind of afterthought near the paper's end.

2) Some further clarifications might be helpful for the introductory paragraphs:

a. Near the start of the paper, the authors first note “As with all complex physiological processes, genetic effects likely play a role, but their extent and their molecular basis are largely unknown” (lines 35-37) which is a fair statement to make. But they then go on to say soon after that “it is unclear to what extent which, if any, phonation characteristics are learnt rather than inherited” (lines 41-42). This latter statement seems to adopt a less nuanced perspective than the earlier statement, implying a dichotomous view in which each phonation characteristic will ultimately be classified as either “learnt” or “inherited”. The biological reality is likely to involve a complex mixture of genetic and environmental factors for the range of phenotypes, rather than a separation of different features into two boxes of “learnt” versus “inherited”.

b. The authors acknowledge “the identification of mutations in single genes that cause speech disorders” but follow this with “Far fewer studies have looked at the genetic basis of speech production” (lines 40-41). I find this a bit confusing. The relevant monogenic disorders mainly include those primarily characterized by deficits in sequencing the orofacial/mouth movements involved in speech (childhood

apraxia of speech), so it seems odd to say that those studies did not “look at the genetic basis of speech production”. Most probably, I am misunderstanding the intended point of the authors here. Perhaps they were hoping to highlight interindividual variation in speech production in the general population beyond disorder? Or non-neural effects on speech production? Some rephrasing could help clarify.

c. “Finding the molecular basis of speech production could shed light on this key, uniquely human, attribute.” Although speech is (to the best of our knowledge) unique to humans, the phenotype being focused on in the present study, that of an individual’s median F0 (albeit assessed during speech), may not necessarily have that much to tell us about human-specific aspects. Many non-human animals make vocalizations via laryngeal mechanisms, and the biological underpinnings of F0 production could be well-conserved across species. Other features of speech production, including modulation of pitch and rapid coordinated changes in vocal tract shape, all under voluntary control, might be more informative in this regard, since they appear to be more distinctive in our species.

3) In presenting/discussing findings for rs11046212 in ABCC9, the authors say that it “was the one most strongly associated with pitch in both samples” (lines 26-27, referring to the current study sample and the prior GWAS in independent Icelandic sample) and “was the strongest associations in our study” (lines 83-84). Looking at Figure 1, and the information given in the text, it seems that the second locus on chromosome 12, centered on SNP rs10859172, was actually the most significant association of the GWAS in the Chinese sample. The magnitude of the beta for rs11046212 is slightly higher than that for rs10859172; perhaps that’s why the authors refer to it as “the one most strongly associated” or “strongest”, meaning they are somehow judging in terms of effect sizes? But given that neither rs11046212 nor rs10859172 actually passes the genome-wide significance threshold in the Chinese sample, I’m not sure how much to make from comparing the effect sizes of the two.

4) The authors argue in this paper that median F0 is an oligogenic rather than polygenic phenotype (i.e. that its genetic architecture will be explained by only a few loci). One motivation behind this argument is that the original Icelandic GWAS identified one prominent association, with an effect size that is larger than typically seen for most complex traits. Considering the magnitude of the sample sizes studied thus far, it might be worth some caution before drawing conclusions in this regard. SNP-heritability of median F0 in the Icelandic sample (up to 13,000 participants) was around 17% (and in the new Chinese sample this is 20%). So even after accounting for ABCC9 effects, there remains plenty of room for polygenic effects elsewhere in the genome, which could be revealed by larger samples with enhanced power. Indeed, the meta-GWAS across the Chinese and Icelandic samples already shows three genome-wide significant loci in addition to the ABCC9 association. Even larger samples would likely reveal additional loci, as seen in history of analyses of other complex traits, and at this stage it seems difficult to predict how many will emerge. The authors argue that their PGS results also support oligogenicity: “the best fitting PGS from the Iceland study to explain pitch in the Chinese sample uses only 5 SNPs, suggesting a similar oligogenic (rather than polygenic) architecture of pitch in China and Iceland” (lines 143-144). But I’m not sure if one can draw conclusions about oligogenicity from this particular analysis, given that even this 5-SNP PGS from Icelandic study accounts for such a small amount of variation in the Chinese cohort (see also point 5 below). Perhaps the authors could offer some more formal analyses of the oligogenicity/polygenicity hypotheses, or alternatively be a bit more circumspect about conclusions at present time?

5) One of the main conclusions of the authors is that their work shows “common genetic underpinnings in Mandarin Chinese (a language in which word meaning is conveyed by variation in pitch) and Icelandic (a language in which pitch does not play this role)” (lines 147-149). Notwithstanding the earlier points about the informativeness of median F0 as a trait, I think perhaps this overstates the commonalities that are supported so far. It is true that the convergence of the ABCC9 findings is very striking, but (as above) the magnitudes of the SNP-heritabilities in both samples (17-20%) suggest roles of signals elsewhere in the genome, and the full degree of genetic overlap remains to be determined. Even the best-fitting PGS from the Icelandic study (the one using only 5 SNPs) explained only 0.61% of the variance in median pitch in the Chinese cohort, which doesn't seem that compelling as evidence of broadly common genetic underpinnings for (roughly) the same trait in two samples. Other arguments for commonality focused on the observations that the four loci detected in the Chinese-Icelandic metaGWAS did not show signs of heterogeneity or different direction of effects, but surely a metaGWAS like that is designed to in particular identify these kinds of shared signals, and still the full genetic architecture in either source sample remains to be determined (given power limitations mentioned earlier), leaving room for differential effects. Perhaps some kind of cross-ancestry genetic correlation using the original Icelandic and Chinese GWAS data could also be informative for this question?

6) As in point 5 above, the metaGWAS of Icelandic/Chinese cohorts identified new significantly associated loci. However, the authors don't comment at all on the nature of those signals, where they are located with respect to genomic organisation, whether they point to particular genes, what links to biological pathways might be emerging, if any, etc. (no ZoomLocus plots and no discussion provided). Are there any connections to the ABCC9 findings, based on what is already known about that gene? It would be helpful for readers to hear more about these loci, even if no clear patterns are emerging.

7) It could be good to add some consideration of potential limitations of the current study related to the fact that it involves a female-only cohort, also taking into account what we know about sex differences related to voice characteristics from prior work (including the Icelandic study).

Response to Reviewer #1

1. Phenotypically there is a strong association between MDD status and F0 voice pitch, almost as strong as the association with age, which makes one wonder if voice pitch has a value in predicting MDD. I later realized that in fact, that was a major goal of the study and the authors would want to report that in detail elsewhere as a separate paper. Fair enough. But, as a reader, I'd still like to know what the genetic correlation between MDD and F0 voice pitch is in the current study. Could the authors please report that here? Also, if possible, report the genetic correlation of voice pitch between cases and controls and heritability estimates separately in cases and controls. Finally, polygenic score associations with voice pitch separately in cases and controls and if there are any significant differences in the effect size between the two.

The median pitch, when not adjusted for covariates, was indeed significantly associated with MDD (raw association: $\beta=0.35$, $P=3.68 \times 10^{-56}$). However, unraveling the association between pitch and MDD is a complicated inquiry, which leads to a separate study (the manuscript was enclosed with this one for reference). In short, because the voice signal is sensitive to locations, experimental settings, and environments, we used a meta-analysis of results from each collection site to estimate the association between MDD with pitch-related features. We found a weak association between median pitch and MDD: $\text{Beta}=0.143$, $\text{SE}=0.055$, $P_{\text{FDR}} = 0.013$, but this was not replicated in an independent cohort of 1,084 subjects ($\text{Beta}=0.000$, $\text{SE}=0.126$, $P_{\text{FDR}}=0.997$). The weak phenotype association between pitch and MDD prevented us from further reporting a genetic association. The story is complicated, and requires detailed treatment, but we have included this sentence in the revised manuscript: "Pitch was.... after correcting for the effects of collection site, weakly associated with MDD $\beta =0.14$, $P = 0.013$ "

We reported the SNP-based heritability estimated by LDAK separately in cases and controls in Table S7. In addition, we report the PRS separately in cases and controls (see accompanying figures below). The PRS model explained a proportion of variance in the case/control subgroups that was similar to that in the entire group.

To estimate the genetic correlation of pitch in cases and controls, we used the bivariate GREML method implemented in GCTA. The genetic correlation of pitch between MDD cases and controls was not significant from 1 ($rg=1.00$, $\text{SE}=0.43$, $P=0.5$). It's not clear how useful this result is (it's not significant and it makes some assumptions about the environmental contribution). As an alternative, we tried Popcorn. Looking at the correlation of the same trait in different populations gives a measure of the homogeneity of the trait across the populations. In this case, by construction, only one population-specific measure is observed for each individual. Popcorn was developed to allow the estimation of these cross-population summary statistics measures. However, for reasons discussed in our response to point 2, the measure derived from Popcorn is unreliable.

We've added these new results in the revised manuscript.

Figure S1. The predictive performance of polygenic scores on MDD cases. The number of SNPs is labeled on each bar.

Figure S2. The predictive performance of polygenic scores on controls. The number of SNPs is labeled on each bar.

2. I am curious about the genetic correlation between CONVERGE and deCODE studies. Did the author try to estimate it?

We did try to estimate it, but the estimate is not reliable. We used Popcorn to estimate the trans-ancestry genetic correlation. Popcorn analysis requires two reference panels that are representative of the genetic architecture of the two populations. This is crucial for accurately estimating LD patterns for each population to account for the differences in how genetic variants are inherited in each group. However, the CONVERGE genetic data used ow-coverage whole-genome sequencing data. After merging with the 1,190,321 SNPs in the EAS reference panel, only 683,954 SNPs remained. The significant reduction of matching SNPs resulted in an inaccurate estimation of genetic effect correlation (ρ_g) with very large standard errors: $\rho_g = 0.69$, $se = 1.02$, the genetic correlation was indistinguishable from 1 ($P = 0.76$).

Due to these limitations, we did not report the ρ_g in the manuscript. Instead, we performed a binomial sign test to test whether the genetic effects showed the same direction in CONVERGE and deCODE. In the revised manuscript, we now report the results of the binomial sign test:

“We evaluated whether the observed fraction of results displaying the same direction of allelic effects across studies was significantly greater than expected by chance (that is, 50%) using binomial sign tests. Table 1 gives the number of LD-independent SNPs in the Iceland study at a set of P-value thresholds, and the fraction of these SNPs displaying the same direction of effect in the Chinese group and a one-sided binomial test P-value. 98.99% (97 of 98) of the SNPs showed the same effect direction in the two studies at the P-value threshold $<1 \times 10^{-6}$ (Binomial $P = 1.58 \times 10^{-28}$). Similarly significant binomial test p-values at other P_thresholds.”

Table 1. Binomial Sign Test. For varying P-value thresholds (P_threshold), total the number of SNPs (N_total), number of SNPs showing consistent direction of effect (N_positive), fraction of these with consistent direction of effect, and significance of a one-sided binomial test (Binomial_P).

P_threshold	N_total	N_positive	fraction	Binomial_P
5.00E-08	20	20	100.00%	9.54E-07
1.00E-07	21	21	100.00%	4.77E-07
1.00E-06	99	98	98.99%	1.58E-28
1.00E-05	133	120	90.23%	3.67E-23

3. The authors use a clumping and thresholding approach to analyze polygenic scores. Did they try using any newer methods such as LDpred or PRSCS? Just because the PRS based on the top variants showed the strongest association, I wouldn't conclude

that the voice pitch phenotype is oligogenic. How much proportion of the heritability is explained by the top variants? Does that number justify the claim that voice pitch is an oligogenic trait?

In response to your comment, we tried LDpred2, in which the Iceland-PRS achieved only 0.011% of R² in predicting pitch in Chinese. We followed the LDpred2 authors' recommendation of restricting the analysis to only the HapMap3 SNPs, which resulted in only 49,558 variants matched for the PRS model. This number of variants is much less than the clumping and thresholding approach (243,136 variants). Thus, the predictive R² for LDpred2 is much lower than for the approach we used (which explained 0.61% of the variance). We decided not to report the LDpred result in the main text.

However, overall we agree that the PRS results are not strong enough to make the conclusion of oligogenicity. In the revised manuscript, we have removed this statement in the Discussion.

4. Did the authors perform fine mapping of the ABCC9 locus? How many variants fall within the 95% credible set? Does meta-analyzing Chinese and Icelandic studies help narrow down the causal variants?

Following your suggestion, we performed fine-mapping analysis of the ABCC9 locus, which reduced the number of variants as probable causal variants in the 95% credible set from 15 to 4. The results are listed in Table S6. The variant rs11046212 had the maximum posterior probability (0.35), which was inferred as causal in both EUR and EAS populations (population-specific causal probability >0.99).

5. The authors avoided reporting the genetic correlation of voice pitch with other diseases and traits. Is there any specific reason? I'd assume that meta-analysis would provide more power to do such analysis, and the results would provide some interesting new insights into the environment and disease correlates of voice pitch in the population.

We don't report genetic correlations because the results are not robust. We need to use a method that works on summary statistics, such as LDSC, and here we run into problems because of low power. Our sample size is relatively small, and the methods require a reference panel representative of the genetic architecture of the sample, which we lack. LD Score regression (LDSC) and similar methods for estimating genetic correlation rely heavily on population-specific linkage disequilibrium (LD) patterns. The mixed ancestry in our meta-analysis complicates the use of these methods, as they assume a homogeneous population structure. Applying LDSC to our combined European and East Asian data could lead to biased genetic correlation estimates due to the variation in LD patterns between these populations. Thus, we do not analyze the genetic correlation based on the meta-GWAS results. In future work we will try to circumvent these problems to explore such genetic correlations.

6. Did the authors explore stratified heritability analysis to identify tissues and cell types associated with voice pitch? With a heritability of 20% using LDSC, I'd assume that the sumstats would be well suited for such an analysis.

Using LDAK, the SNP-based heritability is 20%. However, for the same reason as in Comment 5, we do not report the LDSC analysis in our data: LDSC produced an estimation of heritability that was low, with relatively large standard errors (LDSC $h^2 = 0.0368$, SE = 0.0651, Mean $\chi^2 = 1.0045$). We expect stratified analyses using LDSC to be even more underpowered, and even LDAK analyses to be underpowered and uninformative for these analyses and for these analyses to be further limited by lack of annotated SNP sets specific to the East Asian study participants in the current study.

For comparison, we analyzed data from the Iceland study, which was better suited for LDSC due to its larger sample size and higher coverage. The results from the Iceland data indicated a higher heritability estimate with a smaller standard error (LDSC $h^2 = 0.1579$, SE = 0.0384, Mean $\chi^2 = 1.0374$), and a higher proportion of SNPs matched with the European (EUR) reference panel (1,138,858 out of 1,190,321 SNPs). This contrast further illustrates the limitations faced with the CONVERGE data in conducting a stratified heritability analysis.

7. The methods section doesn't describe anything about the genetic data of CONVERGE participants. It is important to add that information here, even though it may have been described elsewhere. Will the summary statistics made available publicly?

We have added more description about the genetic data of CONVERGE in the revised manuscript: "DNA was extracted from saliva samples using the Oragene protocol. Genotypes were acquired from low-coverage sequencing data from which SNPs were imputed. A sensitivity threshold of 90% to SNPs in the 1000G Phase1 ASN panel was applied for SNP selection for imputation. Genotype likelihoods were calculated using a sample-specific binomial mixture model implemented in SNPtools (version 1.0)²⁵, and imputation was performed using BEAGLE (version 3.3.2)²⁶. A second round of imputation was performed with BEAGLE at biallelic SNPs polymorphic in the 1000G Phase 1 ASN panel using the 1000G Phase 1 ASN haplotypes as a reference panel. A final set of allele dosages and genotype probabilities was generated from these two datasets by replacing the results in the former with those in the latter at all sites imputed in the latter. We applied a conservative set of inclusion threshold for SNPs for genome-wide association study: a) p-value for violation Hardy Weinberg equilibrium $p > 10^{-6}$, b) Information score $p > 0.9$, c) minor allele frequency $> 0.5\%$. Full details of the method, and results are given in²²."

We've made our GWAS summary statistics publicly available at FigShare (10.6084/m9.figshare.24995963)

Response to Reviewer #2

However, I do have reservations regarding the framing of the study, especially with respect to what the study can (or rather cannot) reveal about tonal/non-tonal language distinctions, and the biology of human speech more broadly, and would recommend more caution in the conclusions that the authors are drawing.

In response to your feedback, we have carefully revised the abstract, introduction, and conclusion of our manuscript to better align with the actual findings of our study, ensuring that it accurately represents the scope of our research without overstating the implications.

1) The overarching framing of the study could be misleading (through the abstract and in several places in the introductory paragraphs of the paper) with potential to fuel misunderstandings by readers, especially for those who may be less familiar with speech/language sciences.

a. The abstract sets out the apparent scope of the study with the following: "The origins of tonal and non-tonal languages have long been a subject of linguistic inquiry. In tonal languages, such as Mandarin Chinese, pitch changes differentiate word meanings, whereas in non-tonal languages, such as Icelandic, pitch is used to convey intonation. We addressed this question by searching for genetic associations with variation in pitch..." (lines 21-24). Crucially, the phenotypic measure that is under investigation in the current study is the median fundamental (F0) frequency of an individual's voice i.e. the median natural pitch that a person uses when they are speaking. But the role of tone in linguistics concerns how a speaker uses pitch modulation within an utterance in relation to prosody (as in non-tonal languages) or to directly modify the meaning/inflection of words (as in tonal languages). A person's median F0, while certainly an interesting trait to investigate in its own right, is not so likely to be informative for resolving long-standing questions about "origins of tonal and non-tonal languages". A more meaningful target phenotype for a genetic study related to tonal/non-tonal distinctions would be a measure of a person's ability to modulate tone around their median pitch; notably, this avenue is not pursued within the current paper.

We agree with your assessment that investigating the median fundamental frequency (F0) of voice, while valuable, may not directly address the complex questions surrounding the origins of tonal and non-tonal languages. The role of tone in linguistics, as you point out, involves pitch modulation and its use in modifying word meaning in tonal languages or conveying prosody in non-tonal languages. Our study's focus on the median F0 serves as an initial exploration into the genetic basis of a voice characteristic that, while not directly related to linguistic tone modulation, could provide foundational insights into the broader genetic influences on vocal pitch.

Therefore, in light of your feedback, we have revised the abstract to better reflect the specific scope of our study and to avoid any overstatement of its implications. The revised abstract now reads: "The genetic influence on human vocal pitch in tonal and non-tonal languages remains largely unknown...We addressed this question by searching for genetic associations with interindividual variations in median pitch in a Chinese major depression case-control cohort and compared our results with a genome-wide association study from Iceland."

This revision aims to clarify that our investigation is a starting point in exploring the genetic aspects of human vocal pitch, rather than a direct exploration of the linguistic origins of tonal and non-tonal languages.

b. The above issue is compounded by some lack of clarity/precision in phrases like "genetic associations with variation in pitch" (line 24) and "the first genetic locus associated with variation in voice pitch" (lines 44-45). Such phrases do not make clear that "variation" here is meant as a group-level reference to interindividual variation of median pitch rather than a claim about identifying genetic associations with a phenotype based on intraindividual variation (i.e. how tone is being modulated during utterances by each person in a cohort). It makes it open to misunderstanding by non-expert readers, who are likely to assume the latter. And nowhere in the title or abstract do the authors mention that the focus here is limited to a person's median pitch, which makes it even more likely that readers could (given the predominant tonal/non-tonal framing) jump to the wrong conclusions about the nature of the target trait.

We've revised these phrases to avoid such confusion and made it clear in the abstract that the focus here is a person's median pitch:

We have revised the title to read 'Genetic association analysis of human median voice pitch...'

In the Abstract, we revised the original text as "We addressed this question by searching for genetic associations with interindividual variation in median pitch..."

In Introduction, we revised the original text as "A recent genome-wide association study (GWAS) discovered the first genetic locus associated with median voice pitch..."

c. In the final sentence of the abstract, the authors conclude that their findings show "a genetic contribution to a fundamental capability, the physiological basis for pitch control in speech, shared by all humans, regardless of their linguistic or cultural background" (lines 29-31). Given the restricted focus on interindividual variation in median F0, GWAS information from only two languages/populations, and without any investigations of how genetic factors influence the modulation of tone while speaking, it seems to be an overstatement to make such a broad claim about shared genetics of "pitch control in speech" across linguistic/cultural backgrounds. As far as I can see, that claim is not something that can be resolved with the current limited study design, and so I would recommend rewriting the abstract with a more measured and modest conclusion. In addition, some adjustment of the paper's title seems warranted to reflect these points and to avoid casual readers being misled over the broader significance of the findings for understanding of tonal/non-tonal distinctions.

We agree and in the revised manuscript, we draw a more modest conclusion: “The discovery of genetic variants influencing vocal pitch across both tonal and non-tonal languages suggests the possibility of a common genetic contribution to the human vocal system shared in two distinct populations with languages that differ in tonality (Icelandic and Mandarin).”

d. To the credit of the authors, very close to the end of the manuscript, they do concede that the chosen trait for study here is not that informative for investigating tonal/non-tonal questions: “Marked differences in pitch patterns between tonal and non-tonal languages have been demonstrated in previous studies yet we found a cross-linguistic consistency in the influence of ABCC9 locus on pitch. We think this consistency is because the analyzed phenotype, median F0, is primarily related to the non-linguistic production of speech, rather than to meaning. Spoken language, which involves cognition and emotional process, is more likely to be reflected in the variation of F0 over time during speech, not to the mean or median values.” (Lines 149-155). I agree, and would suggest that the lack of informativity of median F0 with respect to deeper questions about biology of speech and language (including tonal/non-tonal distinctions) needs to be acknowledged upfront in the manuscript, and should be taken into account for the overall framing of the work, rather than being treated as a kind of afterthought near the paper’s end.

In the revised manuscript, we now acknowledge upfront in the introduction that “There is a broad set of acoustic measures, each representing a specific aspect of the human vocal system and its psychological correlates.” and the current study is based on “A recent genome-wide association study (GWAS) discovered the first genetic locus associated with median voice pitch”.

2) Some further clarifications might be helpful for the introductory paragraphs:
a. Near the start of the paper, the authors first note “As with all complex physiological processes, genetic effects likely play a role, but their extent and their molecular basis are largely unknown” (lines 35-37) which is a fair statement to make. But they then go on to say soon after that “it is unclear to what extent which, if any, phonation characteristics are learnt rather than inherited” (lines 41-42). This latter statement seems to adopt a less nuanced perspective than the earlier statement, implying a dichotomous view in which each phonation characteristic will ultimately be classified as either “learnt” or “inherited”. The biological reality is likely to involve a complex mixture of genetic and environmental factors for the range of phenotypes, rather than a separation of different features into two boxes of “learnt” versus “inherited”.

We do not mean to imply a dichotomous view for the influence of genetics and environment. We have revised this part as “Far fewer studies have looked at the genetic basis of speech production with acoustic measures, and it is unclear to what extent, if any, phonation characteristics are learnt rather than inherited (or influenced by the interplay of both).”

b. The authors acknowledge “the identification of mutations in single genes that cause speech disorders” but follow this with “Far fewer studies have looked at the genetic basis of speech production” (lines 40-41). I find this a bit confusing. The relevant monogenic disorders mainly include those primarily characterized by deficits in sequencing the orofacial/mouth movements involved in speech (childhood apraxia of speech), so it seems odd to say that those studies did not “look at the genetic basis of speech production”. Most probably, I am misunderstanding the intended point of the authors here. Perhaps they were hoping to highlight interindividual variation in speech production in the general population beyond disorder? Or non-neural effects on speech production? Some rephrasing could help clarify.

Our reference to "speech production" meant vocal characteristics in speech, quantified by acoustic measures, such as pitch and intonation, in the general population. We've rephrased this sentence as "Far fewer studies have looked at the genetic basis of speech production with acoustic measures"

c. “Finding the molecular basis of speech production could shed light on this key, uniquely human, attribute.” Although speech is (to the best of our knowledge) unique to humans, the phenotype being focused on in the present study, that of an individual's median F0 (albeit assessed during speech), may not necessarily have that much to tell us about human-specific aspects. Many non-human animals make vocalizations via laryngeal mechanisms, and the biological underpinnings of F0 production could be well-conserved across species. Other features of speech production, including modulation of pitch and rapid coordinated changes in vocal tract shape, all under voluntary control, might be more informative in this regard, since they appear to be more distinctive in our species.

We agree that other features of speech production might be more informative. However, currently we start from the median pitch as it is the only acoustic measure studied in this field. We do not ignore the importance of other kinds of acoustic measures. In contrast, our results highlight the value of studying other features. To clarify, we've revised the original text as “Far fewer studies have looked at the genetic basis of speech production with acoustic measures, and it is unclear to what extent which, if any, phonation characteristics are learnt rather than inherited (or influenced by the interplay of both). Finding the molecular basis of speech acoustics could shed light on this key, uniquely human, attribute.”

3) In presenting/discussing findings for rs11046212 in ABCC9, the authors say that it “was the one most strongly associated with pitch in both samples” (lines 26-27, referring to the current study sample and the prior GWAS in independent Icelandic sample) and “was the strongest associations in our study” (lines 83-84). Looking at Figure 1, and the information given in the text, it seems that the second locus on chromosome 12, centered on SNP rs10859172, was actually the most significant association of the

GWAS in the Chinese sample. The magnitude of the beta for rs11046212 is slightly higher than that for rs10859172; perhaps that's why the authors refer to it as "the one most strongly associated" or "strongest", meaning they are somehow judging in terms of effect sizes? But given that neither rs11046212 nor rs10859172 actually passes the genome-wide significance threshold in the Chinese sample, I'm not sure how much to make from comparing the effect sizes of the two.

We have revised the text to read "was one of the most strongly associated loci with pitch in both samples".

4) The authors argue in this paper that median F0 is an oligogenic rather than polygenic phenotype (i.e. that its genetic architecture will be explained by only a few loci). One motivation behind this argument is that the original Icelandic GWAS identified one prominent association, with an effect size that is larger than typically seen for most complex traits. Considering the magnitude of the sample sizes studied thus far, it might be worth some caution before drawing conclusions in this regard. SNP-heritability of median F0 in the Icelandic sample (up to 13,000 participants) was around 17% (and in the new Chinese sample this is 20%). So even after accounting for ABCC9 effects, there remains plenty of room for polygenic effects elsewhere in the genome, which could be revealed by larger samples with enhanced power. Indeed, the meta-GWAS across the Chinese and Icelandic samples already shows three genome-wide significant loci in addition to the ABCC9 association. Even larger samples would likely reveal additional loci, as seen in history of analyses of other complex traits, and at this stage it seems difficult to predict how many will emerge. The authors argue that their PGS results also support oligogenicity: "the best fitting PGS from the Iceland study to explain pitch in the Chinese sample uses only 5 SNPs, suggesting a similar oligogenic (rather than polygenic) architecture of pitch in China and Iceland" (lines 143-144). But I'm not sure if one can draw conclusions about oligogenicity from this particular analysis, given that even this 5-SNP PGS from Icelandic study accounts for such a small amount of variation in the Chinese cohort (see also point 5 below). Perhaps the authors could offer some more formal analyses of the oligogenicity/polygenic hypotheses, or alternatively be a bit more circumspect about conclusions at present time?

We realized that the conclusion about oligogenicity is not well substantiated. We've removed this conclusion in the revised manuscript.

5) One of the main conclusions of the authors is that their work shows "common genetic underpinnings in Mandarin Chinese (a language in which word meaning is conveyed by variation in pitch) and Icelandic (a language in which pitch does not play this role)" (lines 147-149). Notwithstanding the earlier points about the informativeness of median F0 as a trait, I think perhaps this overstates the commonalities that are supported so far. It is true that the convergence of the ABCC9 findings is very striking, but (as above) the magnitudes of the SNP-heritabilities in both samples (17-20%) suggest roles of signals elsewhere in the genome, and the full degree of genetic overlap remains to be determined. Even the best-fitting PGS from the Icelandic study (the one using only 5

SNPs) explained only 0.61% of the variance in median pitch in the Chinese cohort, which doesn't seem that compelling as evidence of broadly common genetic underpinnings for (roughly) the same trait in two samples. Other arguments for commonality focused on the observations that the four loci detected in the Chinese-Icelandic metaGWAS did not show signs of heterogeneity or different direction of effects, but surely a metaGWAS like that is designed to in particular identify these kinds of shared signals, and still the full genetic architecture in either source sample remains to be determined (given power limitations mentioned earlier), leaving room for differential effects. Perhaps some kind of cross-ancestry genetic correlation using the original Icelandic and Chinese GWAS data could also be informative for this question?

We tried to use Popcorn to estimate the trans-ancestry genetic correlation. Popcorn analysis requires two reference panels that are representative of the genetic architecture of the two populations. This is crucial for accurately estimating LD patterns for each population to account for the differences in how genetic variants are inherited in each group. However, our CONVERGE dataset comprised low-coverage whole-genome sequencing data. When we merged this with the 1,190,321 SNPs in the 1000 Gnome East Asian (EAS) reference panel, only 683,954 SNPs remained. This significant reduction of SNPs (and limited sample size in CONVERGE) resulted in an inaccurate estimation of genetic effect correlation (ρ_g) with very large standard errors: $\rho_g = 0.69$, $se = 1.02$, the genetic correlation was indistinguishable from 1 ($P = 0.76$). Due to these limitations, we did not report the ρ_g in the manuscript. Instead, we performed a binomial sign test to test whether the genetic effects showed the same direction in CONVERGE and deCODE. In the revised manuscript, we now report the results of the binomial sign test:

“We evaluated whether the observed fraction of results displaying the same direction of allelic effects across studies was significantly greater than expected by chance (that is, 50%) using binomial sign tests. Table 1 gives the number of LD-independent SNPs in the Iceland study at a set of P-value thresholds, and the fraction of these SNPs displaying the same direction of effect in the Chinese group and a one-sided binomial test P-value. 98.99% of the SNPs showed a same direction at a P-value threshold $<1 \times 10^{-6}$ (Binomial $P = 1.58 \times 10^{-28}$).”

Table 1. Binomial Sign Test. For varying P-value thresholds (P_threshold), total the number of SNPs (N_total), number of SNPs showing consistent direction of effect (N_positive), fraction of these with consistent direction of effect, and significance of a one-sided binomial test (Binomial_P).

P_threshold	N_total	N_positive	fraction	Binomial_P
5.00E-08	20	20	100.00%	9.54E-07
1.00E-07	21	21	100.00%	4.77E-07
1.00E-06	99	98	98.99%	1.58E-28
1.00E-05	133	120	90.23%	3.67E-23

6) As in point 5 above, the metaGWAS of Icelandic/Chinese cohorts identified new significantly associated loci. However, the authors don't comment at all on the nature of those signals, where they are located with respect to genomic organisation, whether they point to particular genes, what links to biological pathways might be emerging, if any, etc. (no ZoomLocus plots and no discussion provided). Are there any connections to the ABCC9 findings, based on what is already known about that gene? It would be helpful for readers to hear more about these loci, even if no clear patterns are emerging.

In response to your feedback, we have now included additional information in our manuscript to provide a deeper understanding of the associated loci:

1) Regional plots for GWAS Hits using LocusZoom:

We have updated Figure 1 to include the regional LocusZoom plots for each of the identified loci. This addition aims to offer insights into the potential functional relevance of these genetic associations.

2) Cross-Ancestry Fine-Mapping Analysis at ABCC9 Locus:

We conducted a cross-ancestry fine-mapping analysis specifically focused on variants at the ABCC9 locus. This analysis has allowed us to narrow down the number of probable causal variants in the 95% credible set. We reduced the number of candidate variants from 15 to 4, as detailed in Table S6.

Notably, the variant rs11046212 emerged with the highest posterior probability (0.35) of being the causal variant. This variant was inferred as likely causal in both European (EUR) and East Asian (EAS) populations, with a population-specific causal probability exceeding 0.99.

These enhancements in our manuscript aim to address the gaps in our initial discussion about the genomic organization and biological implications of the identified loci. We believe that these additions will provide the readers with a more comprehensive understanding of our findings and their potential significance in the context of human genetics.

7) It could be good to add some consideration of potential limitations of the current study related to the fact that it involves a female-only cohort, also taking into account what we know about sex differences related to voice characteristics from prior work (including the Icelandic study).

In our revised manuscript, we have included the following limitations: "There are several limitations to our study. First, only women were included in the CONVERGE analysis. Although the Iceland study indicated that the effects of ABCC9 are irrespective of sex, it is unknown whether this finding is applicable to Chinese populations. Second, our analysis focused solely on mapping median F0. Other features, such as the variability of F0 and vowel acoustics, which represent different and possibly more crucial aspects of human vocal control ability, remain unexplored in our study."

REVIEWERS' COMMENTS:

Reviewer #1 (Remarks to the Author):

I thank the authors for addressing my comments and incorporating my suggestions. I have no further comments

Reviewer #2 (Remarks to the Author):

With the revised manuscript, the authors have clearly addressed the issues raised in my prior review.